# Enhancing the thermostability and activity of glycosyltransferase UGT76G1 via computational design

Seong-Ryeong Go [1,2,4], Su-Jin Lee [1,2,4], Woo-Chan Ahn [1], Kwang-Hyun Park [1,2 ✉] & Eui-Jeon Woo [2,3 ✉]

The diterpene glycosyltransferase UGT76G1, derived from *Stevia rebaudiana*, plays a pivotal role in the biosynthesis of rebaudioside A, a natural sugar substitute. Nevertheless, its potential for industrial application is limited by certain enzymatic characteristics, notably thermostability. To enhance the thermostability and enzymatic activity, we employed a computational design strategy, merging stabilizing mutation scanning with a Rosetta-based protein design protocol. Compared to UGT76G1, the designed variant 76_4 exhibited a 9 °C increase in apparent $T_m$, a 2.55-fold increase rebaudioside A production capacity, and a substantial 11% reduction in the undesirable byproduct rebaudioside I. Variant 76_7 also showed a 1.91-fold enhancement rebaudioside A production capacity, which was maintained up to 55 °C, while the wild-type lost most of its activity. These results underscore the efficacy of structure-based design in introducing multiple mutations simultaneously, which significantly improves the enzymatic properties of UGT76G1. This strategy provides a method for the development of efficient, thermostable enzymes for industrial applications.

[1] Critical Diseases Diagnostics Convergence Research Center, Korea Research Institute of Bioscience and Biotechnology (KRIBB), Daejeon 34141, Republic of Korea. [2] Department of Proteome Structural Biology, KRIBB School of Bioscience, University of Science and Technology (UST), Daejeon 34113, Republic of Korea. [3] Disease Target Structure Research Center, Korea Research Institute of Bioscience & Biotechnology (KRIBB), Daejeon 34141, Republic of Korea. [4] These authors contributed equally: Seong-Ryeong Go, Su-Jin Lee. ✉email: ruuua@kribb.re.kr; ejwoo@kribb.re.kr

Glycosylation is an essential process in the biosynthesis of various natural compounds. In plants, UDP-dependent glycosyltransferases (UGTs) are the primary enzymes involved in this process[1]. UGTs belong to a large family of enzymes that transfer sugars from donor molecules to acceptor molecules, forming a variety of bioactive glycosides, such as ginsenosides, steviol glycosides, terpenoids, and flavonoids[2–5]. Steviol glucosides (SGs) are natural sweet-tasting compounds found in *Stevia rebaudiana* leaves[6]. Stevioside (ST), a major SG, is approximately 200 times sweeter than sucrose, but it has a bitter aftertaste because it can bind to both human sweet and bitter taste receptors[7]. In contrast, rebaudioside A (RebA), a derivative of ST, is 250 to 450 times sweeter than sucrose and has a more appealing taste profile, with higher stability[8]. This makes RebA more preferred by consumers and industries. However, RebA only accounts for 2–5% of dry leaves, so various strategies, such as selective breeding of *S. rebaudiana*, have been employed to increase its yield[9]. The biosynthetic pathway of SGs in *S. rebaudiana* has been fully elucidated, and scientific efforts are now focused on bioconverting ST to produce SGs with superior taste profiles, such as RebA, RebD, and RebM[10–12]. This process involves coupling enzymes such as sucrose synthase or glucose-6-phosphatase with UGTs to recycle UDP-glucose (UDPG) in microorganisms such as *E. coli* and yeast[13,14]. Improving the enzymatic properties of UGTs has become a critical objective, which includes enhancing catalytic activity, thermostability, heterologous overexpression, and regioselectivity.

UGT76G1 is an enzyme that catalyzes the conversion of ST to RebA and RebD to RebM, using UDPG as a glucose donor[15]. However, its low expression in *E. coli* and short half-life at temperatures above 50 °C limit its usefulness in industrial applications and harsh environments[16]. Additionally, UGT76G1 has the potential to produce an unwanted byproduct, RebI, when it converts excess RebA[4] (Fig. 1a). Therefore, the regioselectivity of the enzyme is critical for its industrial application. Recent advancements in protein engineering methods, such as the ASCA (activity-based sequence conservative analysis) strategy, have been developed to enhance the catalytic efficiency of UGTs[17]. However, many of these approaches require significant time, labor, and high-throughput screening systems. Previous attempts to improve conversion activity with other SGs have been made, but no mutations have been reported that increase the conversion activity of ST to RebA and thermostability[15,18].

This study utilized a suite of protein engineering methodologies to augment both the thermostability and bioconversion activity of UGT76G1. The bioinformatic analysis pinpointed potential mutations, the impacts of which were assessed by contrasting changes in free energy ($\Delta\Delta G$) with the wild-type (WT) structure. Using the Rosetta design protocol, variants were generated, and their structural models were subsequently scrutinized to ascertain the potential impacts of each mutation on enzymatic attributes. This comprehensive approach effectively culminated in the design of UGT76G1 variants exhibiting superior thermostability, amplified regioselectivity, and heightened conversion efficiency.

## Results

### Computational design of stabilized UGT76G1s.
To identify mutations that could enhance the stability of UGT76G1, we employed bioinformatics analysis to generate an initial pool of mutations. This pool was then filtered based on $\Delta\Delta G$ using the ddg_monomer application, followed by structural analyses[19]. We utilized the Rosetta design protocol to create variants with selected mutations, ultimately selecting ten variants for experimental validation based on their structural similarity to the WT protein and thermodynamic energy values (Fig. 1b).

We first analyzed the diversity among homologous protein sequences using a position-specific score matrix (PSSM) to calculate the likelihood of observing each amino acid in a protein alignment[20]. Positive scores in the PSSM signify positions where corresponding amino acids are more likely conserved. Mutations that stabilize these positions can enhance protein function and stability[21,22]. To obtain additional stabilizing mutations, we consulted a FireProt database and identified further mutations that did not overlap with those from the PSSM[23]. Next, the ddg_monomer method was employed to identify mutations that could enhance the stability of a critical structure involved in RebA production[24]. We explored potential optimal conformations for efficient RebA production by relaxing the UGT76G1 structure while bound to UDP and RebA[25]. From the initial pool, we selected 315 mutations with $\Delta\Delta G$ values less than −0.5 kcal/mol. However, a negative $\Delta\Delta G$ value alone does not guarantee structural stability. Therefore, we confirmed the selected mutations through structural analysis and excluded mutations that did not meet specific criteria, such as unsatisfied hydrogen bonding, exposed hydrophobic surface residues, increased loop flexibility, proline introduction into an alpha-helix, or cavity creation. We also excluded mutations within a ~5 Å radius of active residues, such as His25 or Asp124, to prevent interference with enzyme function. After applying these filters, we obtained a final pool of 115 mutations. To enhance stability prior to design, we removed the flexible 11 residues from its N-terminus that did not contribute to its function. In the design step, we used the Rosetta sequence design protocol, which integrates the PackRotamer and FastRelax algorithms to generate variants with lower thermodynamic energy than the WT structure. The PackRotamer algorithm selects the most favorable amino acid to optimize the side-chain rotamer, while the FastRelax algorithm rapidly perturbs the side chain and backbone to explore the conformational space and minimize energy[26]. By repeatedly applying these algorithms, we generated candidate variants with 10 to 75 mutations (Supplementary Table 1). We screened these variants using root-mean-square deviation (RMSD) analysis for the Cα of each model compared to the WT structure and thermodynamic energy calculations. We selected 10 variants that exhibited energy decreases ranging from 44 to 104 kcal/mol when compared to the WT structure (Fig. 1c).

### Enhanced conversion efficiency of the designed UGT76G1s.
We heterologously expressed WT UGT76G1 and 10 variants (labeled 76_1 to 76_10) in *E. coli*. While nine of the variants were soluble, variant 76_10 was insoluble. The yield of soluble protein from 76_1 to 76_4 variants showed a 10–20% reduction compared to the WT, whereas variants 76_5, 76_6, 76_7, and 76_8 are similar to the WT. To ensure consistency in comparing the glycosyltransferase activity of the WT and variants, we used a maltose-binding protein (MBP) fusion protein for screening. This was necessary because complete cleavage of the MBP tags on variants 76_1, 76_2, and 76_3 with protease was challenging. To evaluate the glycosyltransferase activity of WT and 9 variants, we incubated substrates UDPG and ST with either the WT or designed variants at 37 °C for 18 h and monitored changes in the substrate and product peak with high-performance liquid chromatography (HPLC) (Supplementary Fig. 1a). Linear standard curve of ST, RebA and RebI was obtained in five concentration condition through HPLC analysis for accurate quantitative comparison and we represented concentration of substrate and product above linear curve (Supplementary Fig. 1b–d). We represented the product peak intensity of the WT or three designed variants at

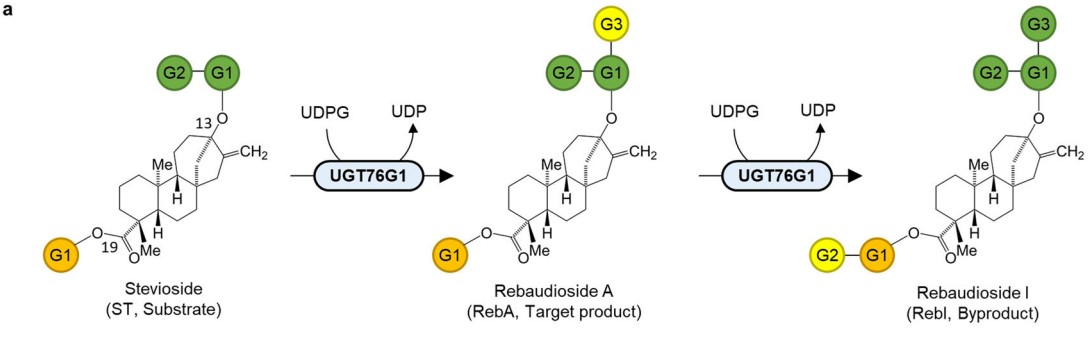

37 °C and the ST, RebA, and RebI peak had retention times of 5.3 min, 4.5 min, and 3.6 min, respectively (Fig. 2a). The production capacity of RebA in WT or the designed three variants was represented as bar graph (Fig. 2b). The highest production capacity of RebA was observed for variant 76_4, which was 2.55 times higher than that of the WT and variant 76_7 which was 1.91 times higher than that of the WT.

Variant 76_4 exhibited elevated activity and this enhancement can be primarily attributed to specific six mutations. Among these, the L101F mutation deserves particular attention due to its significant location (Fig. 2c). In the field of protein design and engineering, enhancing π-π interactions is an essential optimization strategy[27]. These interactions, which are attractive forces between delocalized electrons within aromatic rings, considerably

**Fig. 1 Schematic view of the enzyme reaction and design process. a** The glycosyl transfer pathway mediated by UGT76G1. The glucoses conjugated to steviol aglycon are abbreviated as G1, G2, respectively. The newly attached glucose moieties resulting from the conversion reaction are highlighted in yellow. During the conversion of ST to RebA, UGT76G1 transfers the glucose of UDPG to the first glucose (G1) of C13 in stevioside with a beta-1,3-glycosidic bond. If the glucose of UDPG is transferred to the glucose (G1) of C19 in rebaudioside A, rebaudioside I is produced, which is an unwanted byproduct. **b** A schematic representation of the design strategies to enhance the thermodynamic stability of UGT76G1 through computational design. **c** The mutation profile of the final 10 variants. The components of the N-terminal and C-terminal domains of UGT are shown in the first line. The binding regions for steviol aglycon and glucose moieties are represented as yellow rectangles, while the binding regions for UDPG are displayed as blue rectangles. Catalytic residues (H25, D104) are highlighted in red rectangles. The mutation map for each variant is aligned to the exact location on the domain map, with each mutation represented as a rectangle. The overlapping mutations found in more than 5 variants are labeled with green. Mutations in the substrate binding region are marked with orange. The remaining mutations elsewhere are colored by gray. The number of mutations for each variant is displayed as horizontal bars.

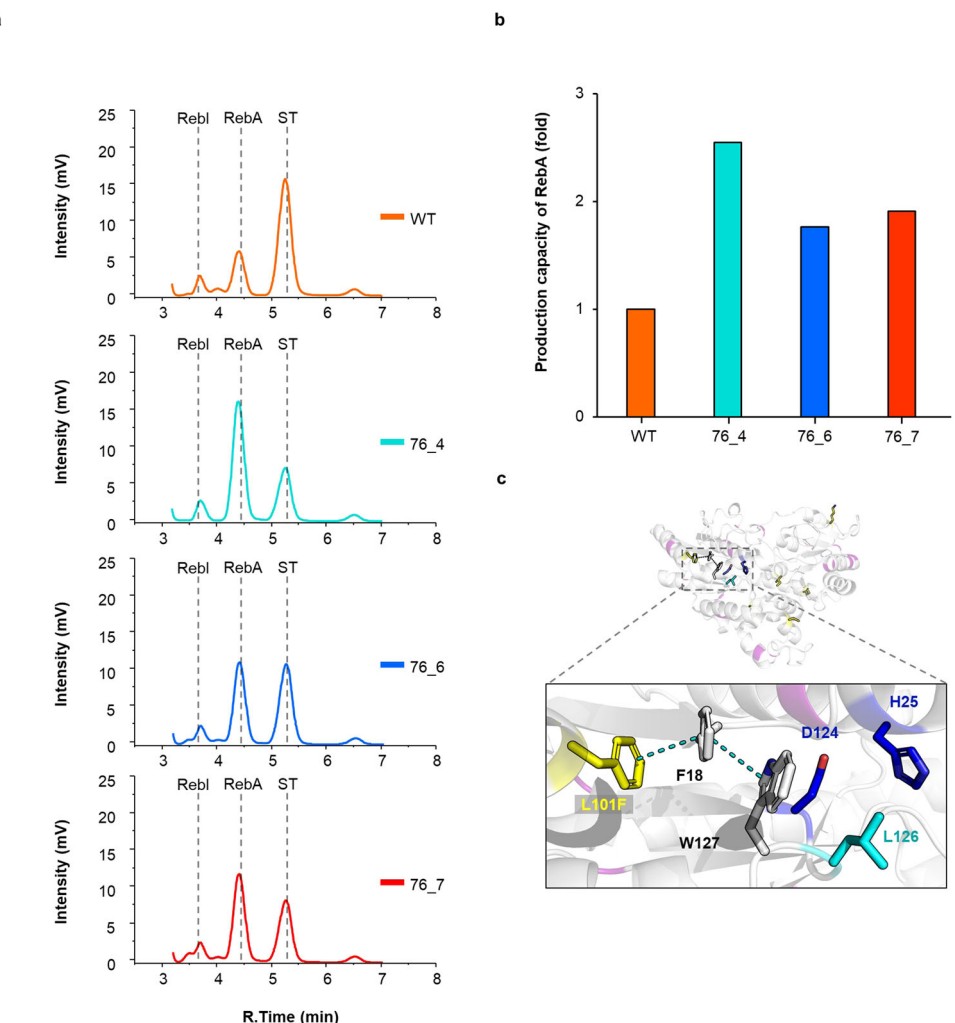

**Fig. 2 Comparison of RebA conversion activity of WT and the designed UGT variants. a** The set of chromatograms representing the activity screening of UGT variants toward ST, RebA, and RebI analyzed using HPLC. The chromatograms include the wild-type (WT, represented in orange), and the variants (76_4 in cyan, 76_6 in blue, 76_7 in red). **b** Production capacity of RebA in WT or the designed variants was represented as a bar graph. The bars include the wild-type (WT, represented in orange), and the variants (76_4 in cyan, 76_6 in blue, 76_7 in red). **c** The Rosetta model structure of 76_4. The mutations overlapping with other variants are highlighted in purple, while the six unique mutations of 76_4 are colored yellow. The cascade of π-π stacking with L101F, F18, and W127 is shown in the box. Catalytic His25 and D124 are represented in blue, and L126, which interacts with the aglycon of ST, is represented in cyan.

boost the stability of protein structures and reduce their overall energy state. In the case of variant 76_4, the L101F mutation triggers a cascade of π-π stacking interactions with the residues Phe18 and Trp127. These residues are strategically positioned near the catalytic residues, such as His25 and Asp124, and the residues that bind directly with the steviol aglycon of stevioside, such as Leu126. The network of π-π interactions instigated by the

L101F mutation in variant 76_4 has the potential to increase catalytic proficiency and specificity toward the substrate, thereby enhancing its conversion efficiency.

**Enhanced thermostability of the designed UGT76G1s.** To evaluate thermal stability, we analyzed the melting temperatures

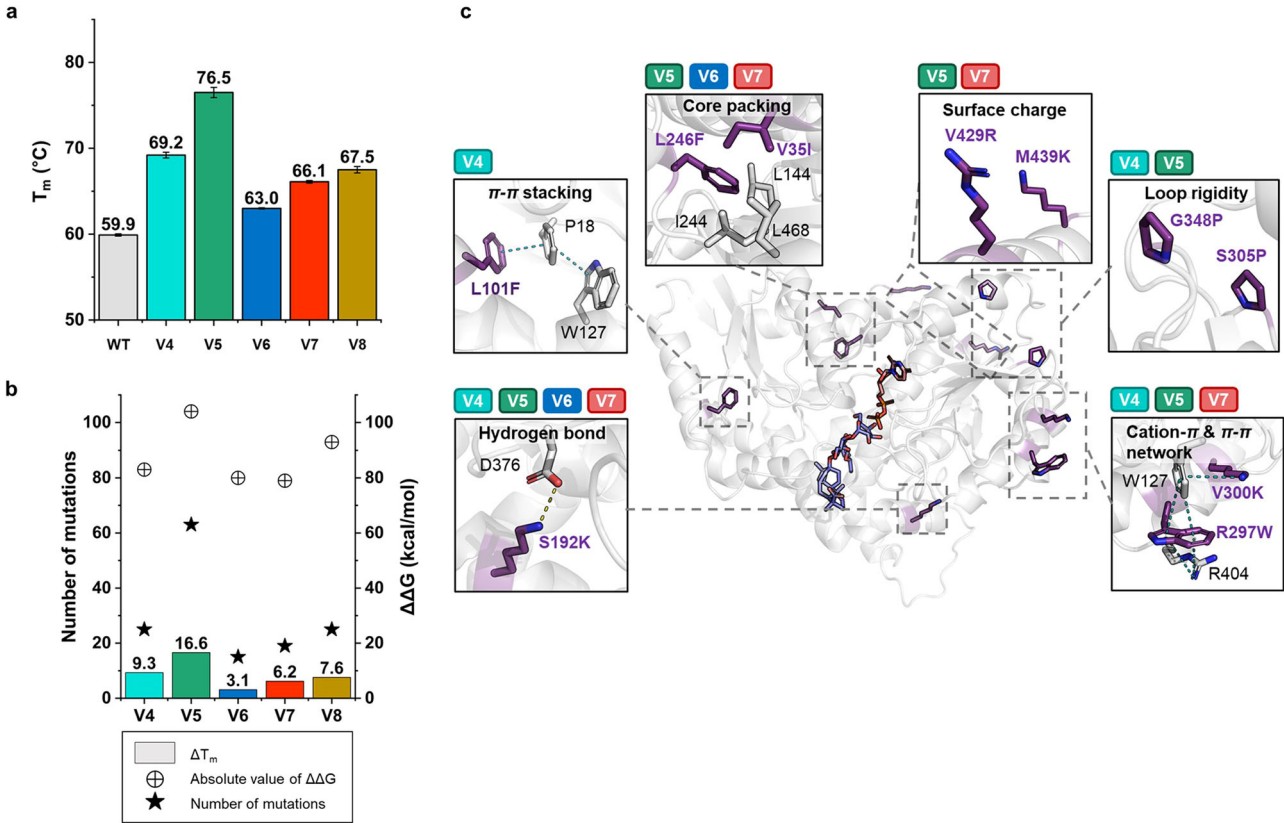

**Fig. 3 Melting temperature of WT and UGT variants and the key stabilizing mutations. a** Melting temperatures ($T_m$) of the wild-type (WT) and five designed variants. Variant 76_4 is abbreviated as V4 (cyan), 76_5 as V5 (bluish green), 76_6 as V6 (blue), 76_7 as V7 (red), and 76_8 as V8 (brown). **b** A comparison of the change in $T_m$ ($\Delta T_m$), the absolute value of $\Delta\Delta G$ (represented by a circle with a cross), and the number of mutations (indicated by a black star). **c** The key mutations that potentially impact the thermostability of the variants. Noncovalent interactions are represented by dashed lines, with hydrogen bonds shown in yellow and $\pi$–$\pi$ or cation–$\pi$ interactions displayed in teal. The mutations present in each variant are indicated above their respective small boxes.

($T_m$) of the WT and five designed variants (76_4, 76_5, 76_6, 76_7, and 76_8) that underwent successful complete MBP tag removal. We used circular dichroism (CD) spectroscopy to determine the $T_m$ values. The WT had a $T_m$ value of 59.9 °C. Interestingly, variant 76_5, which exhibited significantly decreased glycosyltransferase activity compared to the WT, showed a substantial increase in $T_m$, reaching 76.5 °C. This represents a remarkable increase of 16.6 °C compared to the WT. Similarly, variant 76_4, which had a 2.55 times increase in rebaudioside A production capacity than the WT, also displayed a notable rise in $T_m$, measuring 69.2 °C. This corresponds to a significant increase of 9.3 °C relative to the WT. Moreover, variants 76_6, which had a 1.76-fold enhancement rebaudioside A production capacity than the WT, and 76_7, which had 1.91-fold than the WT, revealed moderate yet significant enhancements in $T_m$, measuring at 63.0 °C and 66.1 °C, respectively (Fig. 3a). When the number of mutations introduced into the variant was compared with the amount of change in $T_m$ ($\Delta T_m$) and the stabilized free energy value ($\Delta\Delta G$), it was observed that as the number of mutations increased, a pattern of greater $\Delta\Delta G$ and $\Delta T_m$ values emerged (Fig. 3b).

To study the structural features of mutated forms with higher $T_m$ values, we examined four variant models, all showing improved thermostability. In each of these models, we observed the presence of key residues that are related to structural stability. In variant 76_5, the L246F and V35I mutations are located in the core packing, probably resulting in a more stable assembly among core residues. The S192K mutation formed a new hydrogen bond,

whereas the G348P and S305P mutations likely reduced flexibility in the loop. The V300K and R297W mutations introduced $\pi$–$\pi$ and cation–$\pi$ interactions, respectively, while the V429R and M439K mutations transformed hydrophobic surface residues into ionizable ones, potentially improving solubility and interaction with their aqueous surroundings. Variant 76_4 showed a similar pattern, including the L101F mutation that promoted the formation of $\pi$–$\pi$ stackings, increased loop rigidity by the G348P and S305P mutations, the generation of a new hydrogen bond by the S192K mutation, and the establishment of new $\pi$-$\pi$ and cation-$\pi$ interactions via the V300K and R297W mutations, respectively. Variants 76_6 and 76_7 exhibited mutations that also affected core packing, hydrogen bonding, and $\pi$–$\pi$ stacking, collectively impacting their thermal stability. Taken together, the design method that combined stabilizing mutation scanning with a Rosetta-guided energy calculation introduced multiple mutations simultaneously for enhanced stability (Fig. 3c).

**Enhanced thermal adaptation of the designed UGT76G1s.** A significant number of commercial enzymes, particularly those used in industrial applications, require the ability to function under high-temperature conditions. These thermally resilient enzymes offer several benefits, such as faster reaction kinetics, compatibility with diverse processes, resistance to denaturation, and low risk of microbial contamination due to their high-temperature operation. We investigated the feasibility of using variants for RebA production under high-temperature conditions. To do this, we considered

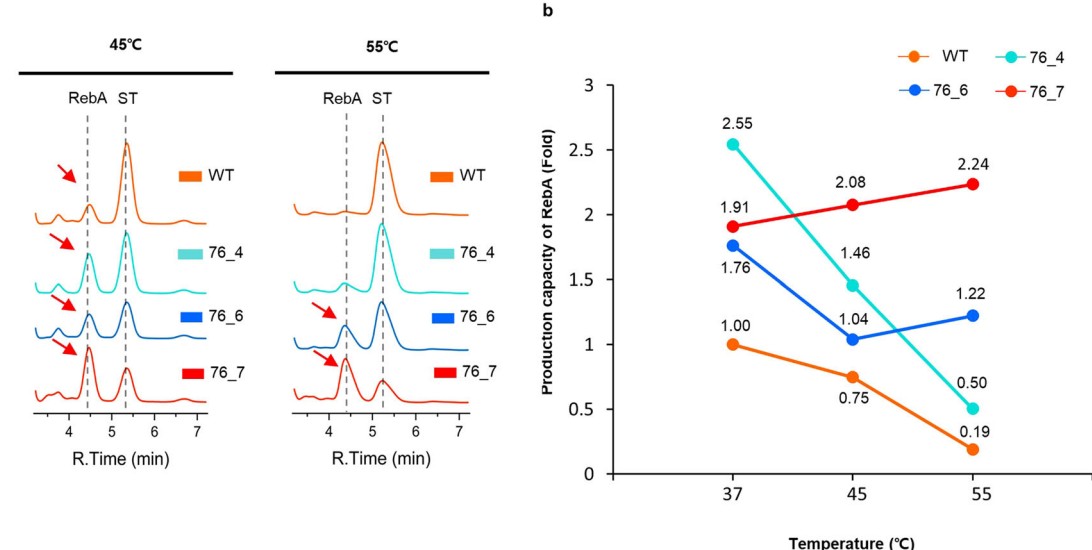

**Fig. 4 Comparison of RebA conversion activity of WT and variants at higher temperatures. a** The chromatogram stack displays the profiles of the wild-type (WT, orange), and variant enzymes (76_4 in cyan, 76_6 in blue, and 76_7 in red). While both WT and 76_4 exhibited reduced activity at 55 °C, 76_6 and 76_7 displayed activity even at high temperatures. **b** Production capacity of RebA of WT, 76_4, 76_6, and 76_7 depends on temperature.

four variants, 76_4, 76_5, 76_6, and 76_7 based on their commendable thermostability and increased $T_m$ values. Variant 76_8, which has relatively highest increased $T_m$ values had no reaction in activity screening of UGT (Supplementary Fig. 1a, b), so we did not select it. When we raised the temperature to 55 °C, we observed a significant reduction in the conversion activity of substrate ST to RebA by the WT (Fig. 4a). Interestingly, at 45 °C, three variants (76_4, 76_6, and 76_7) demonstrated superior conversion activity compared to the WT. However, the variant with the highest $T_m$ value (76_5) failed to exhibit any catalytic activity even at 45 °C (Supplementary Fig. 2a, b).

At 55 °C, despite having the second-highest $T_m$ value, variant 76_4 exhibited a significant decrease in activity. This indicates that the optimal performance of variant 76_4 was limited to temperatures up to 37 °C (Fig. 4b). In contrast, variant 76_6, while exhibiting a reduced conversion activity at 55 °C compared to 37 °C, still performed better than the WT at 55 °C. Remarkably, variant 76_7 demonstrated a production capacity of RebA at 55 °C that was about twice that of the WT and retained its maximal glycosyltransferase activity at 55 °C. This indicates that the introduced mutations not only increased the $T_m$ values but also maintained catalytic activity at high temperatures. Thus, variants 76_6 and 76_7 show promise for application in high-temperature reactions.

**Enhancing regioselectivity and enzyme dynamics of the designed UGT76G1s.** The prolonged reaction of RebA and UGT76G1 is known to lead to the accumulation of RebI, an undesired byproduct that cannot undergo further conversion since UGT76G1 facilitates the conversion of RebA into RebI. Therefore, it is crucial to develop a UGT76G1 variant that exhibits increased regioselectivity to prevent the formation of RebI. To examine this, we analyzed RebI production in both designed variants and the WT. In the HPLC analysis of the ST to RebA conversion process in the WT, a RebI peak emerged at 3.6 min (Supplementary Fig. 1). Additionally, for a more accurate quantitative determination, we substituted the reactant ST with RebA and assessed the RebA to RebI conversion activity using both HPLC and a glycosyltransferase assay kit. Both results indicate a significant decline in RebI production for variants 76_4 and 76_5 compared to the WT. Specifically, the RebI peak in

variant 76_4 was about three-fifths that of the WT, and in variant 76_5, it was only one-fifth of the WT in HPLC analysis (Fig. 5a and Supplementary Fig. 3a). The enzymatic activity could also be monitored by the reduction of the substrate. To assess the reduction of RebA during the conversion to RebI, we compared the difference in the area of the RebA peak between the control and the reactant catalyzed by WT and its variants. Compared to variant 76_5, the area of RebA of WT decreased by 23%, and compared to variant 76_4, the area of WT decreased by 11% (Supplementary Fig. 3b). Furthermore, analysis using a glycosyltransferase assay kit revealed a significant decrease in RebI production in variants 76_1 to 76_5. These variants produced only approximately 10% of the RebI observed in the WT (Fig. 5b). Among all variants, 76_4 exhibited a considerable increase in RebA production while still achieving an 11% reduction in the undesired byproduct RebI. This highlights its enhanced efficacy in minimizing undesirable side reactions.

To understand the elevated conversion activity of ST to RebA and the diminished conversion of RebA to RebI observed in 76_4, we performed an enzyme kinetic assay comparing the WT and 76_4 during the ST to RebA conversion process. The assay results revealed that 76_4 has improved kinetic properties compared to the WT. Specifically, 76_4 exhibited a $V_{max}$ value 2.5-fold greater than that of the WT, indicating a higher reaction speed. Moreover, 76_4 displayed a $K_m$ value that was only 38% of that of the WT, implying that 76_4 had a stronger affinity for the ST substrate and required less of it to reach its maximal reaction rate. The catalytic rate constant ($k_{cat}$) for 76_4 was found to be almost double that of the WT, suggesting that 76_4 was capable of converting two ST molecules into two RebA molecules within the same timeframe (Fig. 5c). Demonstrating its superior efficiency, the ratio of $k_{cat}/K_m$ for 76_4 was about six times greater than that of the WT, indicating its enhanced proficiency in RebA production.

**Discussion**

RebA, recognized as a safe and natural alternative to sucrose, has garnered considerable interest in multiple industries over the past decade due to its FDA status as generally recognized as safe (GRAS)[28]. In this research, we used computational design strategies with the Rosetta suite to engineer variants of

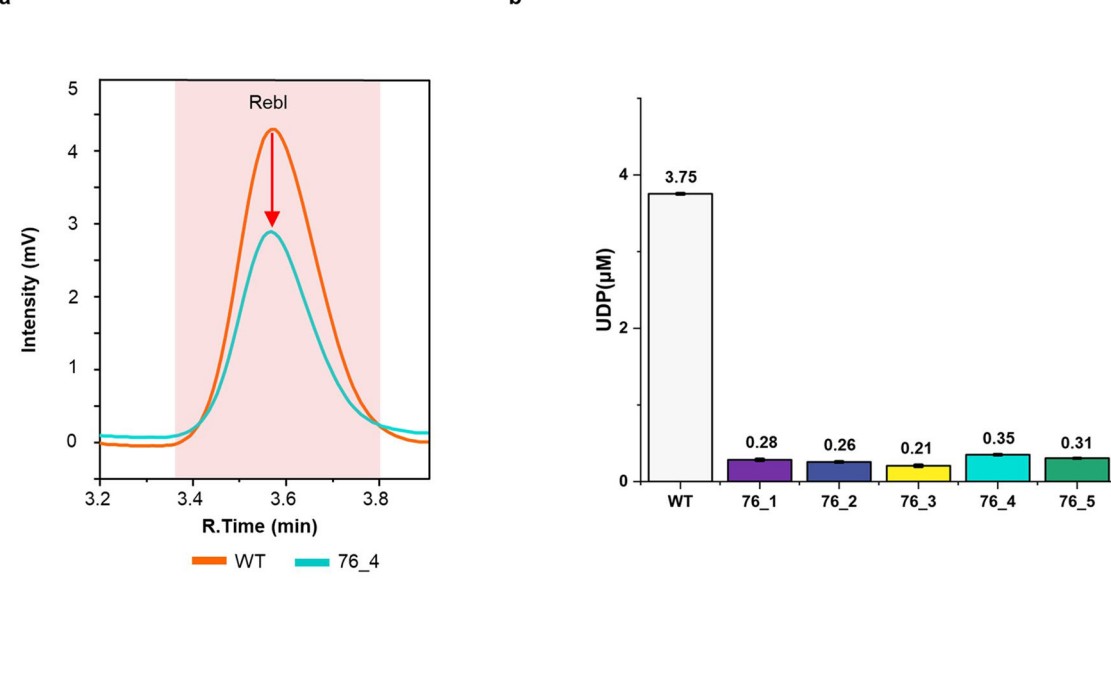

**Fig. 5 Reduction in RebI conversion activity in variants 76_4 compared to WT. a** The chromatogram stack shows the RebI peak of the WT (orange), and variant 76_4 (cyan). **b** Glycosyltransferase assay of the conversion of RebA to RebI. The amount of UDPG used for the conversion of RebA to RebI in the variant enzymes (76_1 as purple, 76_2 as dark blue, 76_3 as yellow, 76_4 as cyan, 76_5 as bluish green) and WT (light gray) was less than 10% of that in WT. The error bars represent the standard deviation, calculated based on duplicated measurements. **c** Kinetic parameters of WT and 76_4 toward ST. The kinetics of WT and 76_4 toward the conversion of ST to RebA were assessed using a glycosyltransferase assay kit. Fold* is the fold change over catalytic efficiency ($k_{cat}/K_m$) of WT.

|  | WT | 76_4 |
|---|---|---|
| $V_{max}$ (µM·min⁻¹) | 2.4 ± 0.2 | 5.9 ± 0.2 |
| $K_m$ (µM) | 92.0 ± 5.3 | 34.5 ± 3.0 |
| $k_{cat}$ (min⁻¹) | 17.3 ± 1.6 | 35.4 ± 1.3 |
| $k_{cat}/K_m$ (min⁻¹·µM⁻¹) | 0.19 ± 0.01 | 1.1 ± 0.1 |
| Fold* | 1.0 | 5.8 |

UGT76G1 that have enhanced features[29]. Out of the 9 analyzed variants, 76_4 was notable for its enhanced ability to convert ST to RebA while minimizing the production of Reb I, an undesirable byproduct. Analysis of the structure suggests that the improved regioselectivity may be due to a stabilization of the ligand within the catalytic pocket. The interaction between the enzyme and substrate can be influenced by the free energy state of the ligand within the binding pocket. To assess this, we used ligand docking simulations with SeeSAR software (Supplementary Fig. 4). The in silico analysis suggested that the estimated binding affinity for ST in the WT fell in the micromolar (µM) to millimolar (mM) range, while the 76_4 variant showed considerably stronger binding, falling in the nanomolar (nM) to µM range (Supplementary Fig. 4c). This coincides with the 34% reduction in $K_m$ value of 76_4 for ST compared to that of the WT, which indicates an increased affinity for ST. This analysis suggests that the improved affinity and regioselectivity of 76_4 toward ST may result from the stabilization of the ligand energy, facilitated by precise modifications within the binding pocket. Thus, we hypothesize that the enhanced affinity of 76_4 for ST may be due to the stabilization of ligand energy, driven by specific distal mutations that lead to structural modifications[30].

Previous studies have observed that designs aimed at increasing the heat resistance of enzymes often decrease enzyme activity[31]. A notable example is variant 76_5, which has 62 mutations resulting in a significant increase of 16.6 degrees in $T_m$ but lost half of its activity, indicating the potential risk of compromising enzyme activity when introducing extensive mutations to strengthen the structure. Interestingly, the variants that demonstrated increased $T_m$ while maintaining activity had approximately 20 to 30 mutations, falling within the stabilization energy range ($\Delta\Delta G$ change) of 20 to 30 kcal/mol. This highlights the importance of fine-tuning and careful adjustment of the design parameters to enhance both thermostability and activity.

The design strategy employed in this study has the potential to enhance the thermostability of similar UGT enzymes, including OsUGT91C1[32,33]. UGT91D2, while capable of converting RebA to RebD in *S. rebaudiana*, suffers from instability and rapid degradation in *E. coli*, resulting in low activity and rendering it unsuitable for industrial applications[10]. Therefore, for industrial conversion of RebA to RebD, OsUGT91C1 (also known as EUGT11) is preferred over UGT91D2 from *Stevia rebaudiana* due to its stability and performance. Given the high sequence similarity and overall structural architecture, employing the same design strategy for this UGT enzyme is worth considering.

Alternatively, combining OsUGT91C1 with the 76_7 variant could establish a comprehensive pathway for industrial RebM production. Introducing a vector encoding these enzymes into yeast holds potential for the large-scale synthesis of steviol glycosides[34]. The 76_7 variant showed a 16% higher conversion activity of RebD to RebM than the WT at 37 °C (Supplementary Fig. 5). Given that the solubility of ST is quite low at room temperature but increases significantly at higher temperatures, we could use a higher concentration of ST substrate at elevated temperatures for its conversion into RebA by variant 76_7.

Furthermore, this design strategy can be extended to engineer other UGT families, including PgUGTs from *Panax ginseng* for ginsenoside production or to other medically significant UGTs involved in the glycosylation of terpenoids, such as bufotalin, neoandrographolide, and flavonoids, to enhance their pharmacokinetics[2,3,35]. The strategy we have outlined could serve as an efficient approach for designing enzymes for bioactive compound production, fostering advancement in both medicinal and industrial biochemistry.

## Methods

**Computational protein design**. To identify potential mutations, we utilized a position-specific scoring matrix (PSSM) derived from the Possum algorithm, which is based on the UniRef50 database with a 0.001 E-value threshold. Using this approach, we narrowed down the list to 1926 residues with a PSSM positive score. Additionally, 11 thermostable mutations that were not found using the PSSM were identified using the FireProt server. To evaluate the impact of single point mutations on the structure of UGT76G1 (PDB ID: 6O88), we used the ddg_monomer application. We highlighted 315 mutations with a negative $\Delta\Delta G$ value, suggesting the potential for protein stabilization. We used PyMOL to analyze and visually inspect the protein structure. We removed mutations that could affect enzyme functionality near the active site, as well as those that did not satisfy hydrogen bonding, had exposed hydrophobic surface residues, enhanced flexibility, inserted a proline into an alpha-helix, created cavities, or had a low-energy contribution to protein stability ($\Delta\Delta G < -0.5$ kcal/mol). We selected 95 candidate residues and used the Rosetta sequence design along with the PackRotamer and FastRelax algorithms to design a low-energy protein structure. We generated 20,000 variant sequences by selecting candidates with 9 to 90 mutations, accounting for 2-20% of the total amino acids in the protein. Finally, we filtered these structures based on their root-mean-square deviation (RMSD) from the wild-type structure and thermodynamic energy values, with reductions in energy ranging.

**Protein expression and purification**. The linear DNA of wild-type UGT76G1 and the 10 variants were de novo synthesized by Integrated DNA Technologies (IDT) and subcloned into the *EcoRI/SalI* site of pMal_Tev_LL vector containing a 10x His tag, MBP tag, long linker, Tev protease cleavage site, and downstream multiple cloning site (MCS). After subcloning, the DNA sequence of the vectors was confirmed by DNA sequencing by Solgent. The expression vector was transformed into *E. coli* DE3 RIL (Agilent) and cultured in 1 L Luria-Bertani (LB) broth supplemented with 100 μg/mL ampicillin at 37 °C until the optical density at 600 nm reached 0.7. Subsequently, protein expression was induced by 1 mM IPTG, and the culture was incubated for 16 h at 18 °C. After centrifugation, the *E. coli* pellets were resuspended in lysis buffer and sonicated, and the lysate was separated into soluble and insoluble fractions by ultraspeed centrifugation. The soluble fraction was subjected to primary purification using a Histrap HP 5 mL column (Cytiva) connected to an AKTA fast protein liquid

chromatography system (AKTA FPLC, Cytiva) for affinity chromatography. To cleave the N-terminal 10x His-MBP tag from the enzymes, the primary purified proteins were dialyzed for 4 h at 4 °C after adding one-tenth of the amount of Tev protease. After dialysis, the proteins were filtered with a 0.45 μm filter and incubated at 4 °C overnight or longer until the ratio of the cleaved band became more than 90%. Secondary purification was carried out using ion exchange chromatography (Resource Q 6 mL, anion exchange, Cytiva) to remove the tag and the protease. The purified enzymes were concentrated using a centrifugal filter, and their buffer was exchanged with Tris-HCl pH 7.5 and 100 mM NaCl. The purified protein was flash-frozen with liquid nitrogen and stored at –80 °C in aliquots.

**Enzyme reaction and HPLC analysis**. The enzyme reaction was carried out with 1 μM UGT, 0.3 mM SGs (ST or RebA), 1 mM UDPG, 100 mM NaCl, and 20 mM Tris-HCl at pH 7.5 for 18 h at 37 °C, 45 °C, and 55 °C in the incubator. The reaction was stopped by adding an equal volume of water-saturated n-butanol, followed by centrifugation and removal of the lower aqueous layer. The upper n-butanol layer was filtered and transferred to an HPLC vial for analysis. HPLC analysis was performed using a Shimadzu UPLC equipped with Sigma–Aldrich Acclaim® Mixed-Mode WAX-1 column. The mobile phase consisted of 100% acetonitrile and 10 mM sodium phosphate buffer at pH 2.7 in a 50:50 (v/v) ratio. At this time, the pH of the sodium phosphate buffer was made by adding phosphoric acid while measuring with an Orion STAR A211 pH meter (Thermo Fisher). The running time of the HPLC program was 25 min with a flow rate of 1.0 mL/min, an oven temperature of 40 °C, a detection wavelength of 210 nm, and an injection volume of 5 μL.

**Analytical method validation**. For the linearity study, each of five levels of ST, RebA, and RebI within the ranges of 25–400 μM were prepared by serial dilution of a 60 mM stock solution in HPLC grade DMSO. AUC (area under the curve) was assessed by analyzing five different concentrations of standard solution, measured on the same day using HPLC. Linear least squares regression was applied to analyze the standard curve of each analyte, and the coefficient of determination ($R^2$) of the regression equation was used to verify linearity. The concentration of substrate and product were calculated based on the standard deviation of the y-intercept and slope of the calibration curve obtained from linear regression.

**Measuring $T_m$ using circular dichroism**. To investigate the thermostability of WT and four designed UGTs (76_4, 76_5, 76_6, and 76_7) that showed improved glycosyltransferase activity, melting temperatures ($T_m$) were measured using circular dichroism (CD) analysis of the intrinsic optical activity at the Korea Basic Science Institute (KBSI). Specifically, WT and the variants were prepared at a concentration of 0.5 mg/mL in 200 μL, and the protein secondary structure changes of the enzyme were measured at 222 nm using a JASCO J-1500 CD measurement device while increasing the temperature from 20 °C to 95 °C at a rate of 0.1 °C every 5 s.

**Glycosyltransferase assay for the conversion of RebA to RebI**. Glycosyltransferase activity of RebA to RebI by WT, 76_4, and 76_5 was measured using the UDP-Glo™ Glycosyltransferase assay kit (Promega)[36]. UDP present in the RebA reactant used in the HPLC experiment was measured using an assay kit, and the UDP concentration in the sample was calculated based on a standard curve for luminescence intensity and concentration of UDP.

**Kinetic assay**. The kinetic properties of both the WT and designed variant (76_4) were compared using the UDP-Glo™ Glycosyltransferase assay kit according to a previous study[37]. The reaction was conducted by varying the concentration of ST from 300 μM to 0 μM while maintaining a final concentration of 1 mM UDPG. To exclude the possibility of conversion of RebA to RebI during the reaction, the enzyme reaction mixture was collected in triplicate at 30 s intervals over a period of 2 min. The luminescence signal of the plates was measured twice using a VICTOR™ X3 multiplate reader, and the signal intensity was converted to UDP concentration using a standard curve. Based on the Michaelis–Menten equation, $k_{cat}/K_m$ was calculated.

**In silico ligand docking**. SeeSAR, software developed by Bio-SolveIT, was used for in silico docking of stevioside to the model structures of WT and 76_4[38]. The Rosetta relaxed model structure was imported into the software, and the residues involved in binding with stevioside were designated as a binding pocket. Next, the structure of rebaudioside A from the existing crystal structure (PDB ID: 6O88) was modified using a molecule editor to create a stevioside ligand structure. In docking mode, standard docking was performed with 500 poses, medium clash tolerance, and consideration of the chair-to-boat form of the ring conformation. From the generated docking structures, a docking model with an affinity of at least the mM unit was selected, followed by visual inspection to determine if stevioside was bound in the C13 or C19 direction and to identify the best affinity range (nM to μM). Furthermore, the affinity of the state bound in the C13 direction was compared to the affinity of the state bound in the C19 direction. Based on the best docking state, template docking was conducted to confirm if there were any additional docking poses with high affinity. The total binding energy of WT and 76_4 was calculated by summing the Hyde scores of all atoms in the ST ligand from each of their best pose[38]. The pictures of the docking pose with Hyde representation were reconstructed by PyMOL software.

## Data availability

The detailed mutation profile of 10 UGT variants is provided with Supplementary information.

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

## Acknowledgements

This work was supported by a Research Program funded by the National Research Council of Science & Technology (NST) [CRC22021-500] and the National Research Foundation of Korea (NRF) [NRF-2021R1A2C2011328, NRF-2021M3A9G8025599, NRF-2022R1C1C1005270, NRF–2021M3H4A1A02051048] and the KRIBB Research Initiative Program [KGM9952314, KGM1392312, KGM5382322].

## Author contributions

The study was conceived by E.-J.W. E.-J.W. and K.-H.P. provided scientific and experimental suggestions. K.-H.P. and E.-J.W. designed the variants. Protein purification was performed by K.-H.P. and S.-R.G., while structural data analysis and refinement were performed by W.-C.A., K.-H.P., and S.-R.G. Biochemical experiments were conducted by S.-J.L. and S.-R.G. Figures created by W.-C.A., K.-H.P., S.-R.G., and S.-J.L. The manuscript was written by K.-H.P., S.-R.G., S.-J.L., and E.-J.W. with input from all authors.

## Competing interests

The authors declare no competing interests.
