## [Peer Review File · Communications Chemistry]

Reviewers' comments:

Reviewer #1 (Remarks to the Author):

Woo, Park and coworkers have reported a structure-based design to improve the enzymatic thermostability of UGT76G1, which catalyzes ST to Reb A. Computational design strategies with the Rosetta suite was used by the authors to obtain variants with improved thermostability. Variants 76_4 and 76_7 were reported to exhibit improved thermostability and enzymatic activity with the investigation of structural features. However, the determination of enzyme activity could not be realized by dividing the area% of product (RebA) by the area% of substrate (ST), leading to the meaningless result on enzyme activity throughout the manuscript. Moreover, there are still some other problems mentioned below need to be solved before its publication. According to the current results, this reviewer believes that the manuscript does not meet the requirements, and is far away from publishing in Communications Chemistry or even any other journals.

1. It is quite puzzling that the authors determine the conversion rate of ST to RebA by dividing the area% of product (RebA) by the area% of substrate (ST) (Page 14, line 22). What is the meaning to compare peak areas of two different compounds with different UV absorption? Also, what is the meaning of the ratio of peak areas for different compounds in pie charts depicted in Figure 2 and extended Fig. 1? Since the authors employed HPLC to test reaction mixtures, standard curves are required at least to quantify the products or substrates. The authors should redetermine the enzyme throughout the manuscript according to the method in reference (Such as Eur Food Res Technol (2023). <https://doi.org/10.1007/s00217-023-04328-4>, etc.).
2. In Fig. 1c, the Num. of mutations is quite hard to understand for readers with blue squares for each mutation. The authors should reconsider how to present Fig. 1c.
3. In Fig. 3a, vertical coordinate should be T_m rather than ΔT_m .
4. In Fig. 5a, it seems that the baseline separation of Reb I was not achieved, especially for 76_4. Therefore, this reviewer wonders whether the data for Reb I is reliable, and recommend the authors to optimize the HPLC conditions to achieve the baseline separation of Reb I for the solid results. And the HPLC spectra of 76_4 in Fig. 5a is not consistent with that in Extended Data Fig. 1a, because the small peak (retention time of 3.3 min) is missing in Extended Data Fig. 1a. The authors should explain for it.
5. Throughout the manuscript, the authors report two promising mutants 76_4 with high selectivity and 76_7 with high reactivity at higher temperatures. However, the study seems not finished to obtain one mutant with both high selectivity and high reactivity at higher temperatures.
6. The authors need to further polish the manuscript, paying more attention to grammar and typos. For examples:
 - 1) Page 6, line 3, rewrite the sentence.
 - 2) Page 7, line 17, "63 °C and 66.1 °C", the decimal places are inconsistent.
 - 3) Page 13, line 20, " $\mu\text{g}\cdot\text{mL}^{-1}$ " rather than " $\mu\text{g ml}^{-1}$ ".
 - 4) Page 15, line 21, 2 min rather than 2 minutes in order to be consistent with the description mentioned above.
 - 5) Page 15, line 24, "kcat/Km" rather than "kcat/Km" (cat and m should be subscript).

...

Reviewer #3 (Remarks to the Author):

This study focuses on employing a Rosetta-based computational design strategy to enhance the thermostability and bioconversion activity of the diterpene glycosyltransferase UGT76G1.

Bioinformatics analysis has been used to generate an initial pool of mutations, and Rosetta ddg_monomer was then performed to screen this library. Ultimately, from this collection, ten prime variants were singled out for experimental validation and several of the designed variants exhibited improved thermostability, conversion efficiency and regioselectivity compared to the wildtype. This work provides a computational approach for the rational design of UGT76G1. It is an interesting study and I recommended the publication after the following points properly addressed:

1. On page 5 line 7, mutations with $\Delta\Delta G$ values less than -0.5 kcal/mol were selected. What was the rationale for choosing this specific cutoff value?
2. Within Figure 1c, the designations for sites N34 and V300 are allocated along the horizontal axis. However, no specific mutants corresponding to these positions are visually represented. Moreover, it is crucial to verify the accuracy of the horizontal axis portrayal in Figure 1c, as it holds the potential to be misleading. For instance, the mutant K52S is situated on the horizontal axis. Yet, in the context of variant information provided in Supplementary Table 1, the K52 position is altered to alanine (A) in four variants (76_2, 76_3, 76_4, and 76_5). Solely relying on Figure 1c could lead to the erroneous inference that K52S is the exclusive mutant present at position 52 within these five variants. This contrasts with the information in Supplementary Table 1. It is essential to meticulously scrutinize Figure 1c to ensure an accurate representation of mutants, aligning it faithfully with the data in Supplementary Table 1. This measure is pivotal to preempt any potential confusion or misinterpretation arising from inconsistencies between the figure and the supplementary table.
3. On page 6, line 21, it is stated that the L101F mutant is an exclusive feature of variant 76_4. Nonetheless, according to Supplementary Table 1, the L101F mutant is also a constituent of variant 76_2.
4. On page 8 line 19, the variant 76_8 exhibits better thermostability and increased T_m values compared to variants 76_6 and 76_7 based on Figure 3a. This raises the question of why variant 76_8 was not selected over 76_6 and 76_7, given its superior thermal properties.
5. On page 16 line 2, the software SeeSAR is mentioned, but no citation is provided.

Overall response to reviewers

We are grateful for the insightful comments provided by the reviewers, which we believe have significantly improved the quality of our manuscript. We have carefully addressed each of the reviewers' comments and have incorporated the requisite modifications into the updated version of the manuscript.

Reviewer #1:

Woo, Park and coworkers have reported a structure-based design to improve the enzymatic thermostability of UGT76G1, which catalyzes ST to Reb A. Computational design strategies with the Rosetta suite was used by the authors to obtain variants with improved thermostability.

Variants 76_4 and 76_7 were reported to exhibit improved thermostability and enzymatic activity with the investigation of structural features. However, the determination of enzyme activity could not be realized by dividing the area% of product (RebA) by the area% of substrate (ST), leading to the meaningless result on enzyme activity throughout the manuscript. Moreover, there are still some other problems mentioned below need to be solved before its publication. According to the current results, this reviewer believes that the manuscript does not meet the requirements, and is far away from publishing in Communications Chemistry or even any other journals.

Question 1

It is quite puzzling that the authors determine the conversion rate of ST to RebA by dividing the area% of product (RebA) by the area% of substrate (ST) (Page 14, line 22). What is the meaning to compare peak areas of two different compounds with different UV absorption? Also, what is the meaning of the ratio of peak areas for different compounds in pie charts depicted in Figure 2 and extended Fig. 1? Since the authors employed HPLC to test reaction mixtures, standard curves are required at least to quantify the products or substrates. The authors should redetermine the enzyme throughout the manuscript according to the method in reference (Such as Eur Food Res Technol (2023). <https://doi.org/10.1007/s00217-023-04328-4>, etc.).

Response:

We sincerely appreciate the critical question you raised regarding the methodology employed for determining the conversion rate of ST to RebA. We acknowledge that we made a significant mistake in our calculation for the enzyme activity assay. During the initial preliminary screening, which was not included in the manuscript, we used the method of calculating the area under the curve (AUC) to compare relative activities, primarily for rough estimation purposes in order to identify mutants of interest. However, as you pointed out, comparing the peak areas of two different compounds with varying UV absorption properties is not an appropriate approach for accurate enzyme activity calculation. We deeply regret this oversight.

In response to your feedback, we have rectified this issue. We have introduced standard curves to precisely quantify both the substrate and the product. These standard curves were constructed using known concentrations of the compounds, along with their respective peak areas obtained through HPLC analysis. We have recalculated the conversion rate, which now provide a more accurate and reliable quantification of the enzyme activity.

We have thoroughly revised the manuscript to reflect this methodology, aligning it with the method cited in the suggested article in Eur Food Res Technol (2023).

The detailed results and data somewhat differ from the previous manuscript; however, the overall result shows the same pattern, supporting the significant impact of the design of this study.

The specific changes, including the presentation of the new standard curves and the recalculated

conversion rates, have been incorporated into Page 6, Lines 7-17 of the revised manuscript.

Once again, we would like to extend our gratitude for your valuable input, which has undeniably improved the quality and scientific rigor of our work.

Question 2

In Fig. 1c, the Num. of mutations is quite hard to understand for readers with blue squares for each mutation. The authors should reconsider how to present Fig. 1c.

Response:

In response to the reviewer's feedback, we have made substantial changes to the presentation of Figure 1c. In the revised figure, we have indicated the mutation sites within the two main domains of the protein.

We believe that these changes will provide readers with a clearer and more easily understandable depiction of the mutation occurrences.

Question 3

In Fig. 3a, vertical coordinate should be T_m rather than ΔT_m .

Response:

The vertical coordinate should indeed represent the melting temperature (T_m) rather than the change in melting temperature (ΔT_m). We understand that incorrect labeling could lead to misinterpretation of the data. To rectify this, we have conducted a thorough review of the manuscript for similar errors and have amended the label in Figure 3a to accurately display ' T_m ' as the vertical coordinate.

Question 4

In Fig. 5a, it seems that the baseline separation of Reb I was not achieved, especially for 76_4. Therefore, this reviewer wonders whether the data for Reb I is reliable, and recommend the authors to optimize the HPLC conditions to achieve the baseline separation of Reb I for the solid results. And the HPLC spectra of 76_4 in Fig. 5a is not consistent with that in Extended Data Fig. 1a, because the small peak (retention time of 3.3 min) is missing in Extended Data Fig. 1a. The authors should explain for it.

Response:

According to the suggestion by the reviewer, we performed additional experiments by optimizing the HPLC conditions in variables such as column temperature, flow rate, and gradient. As a result, we have achieved a clean baseline separation for Reb I. These new data are integrated into Figure 5a, Extended Data Fig. 3, and detailed in Page 9, Lines 15-20 of the revised manuscript. Now, we do not see any discrepancy that was initially observed in the first manuscript and thus pointed out by the reviewer. we appreciate very much for this improvement.

In regard to the discrepancy between 'Fig. 5a' and 'Extended Data Fig. 1a', it's important to clarify that these represent distinct datasets. 'Figure 5a' depicts the conversion activity from RebA to Reb I, with

RebA as the initial substrate, while 'Extended Data Fig. 1a' illustrates glycosyltransferase activity from ST to RebA, with ST as the starting substrate.

Question 5

Throughout the manuscript, the authors report two promising mutants 76_4 with high selectivity and 76_7 with high reactivity at higher temperatures. However, the study seems not finished to obtain one mutant with both high selectivity and high reactivity at higher temperatures.

Response:

Compared to the WT, mutant 76_4 demonstrated an impressive 2.55-fold increase in RebA production capacity at 37°C, accompanied by a notable 9°C increase in apparent T_m . These improvements translated to a 1.46-fold increase in activity at 45°C. Furthermore, this mutant exhibited a substantial 11% reduction in the formation of the undesirable byproduct rebaudioside I. As a result, we firmly believe that mutant 76_4 has successfully achieved the crucial combination of enhanced activity, heightened stability, and improved selectivity.

On the other hand, while mutant 76_7 did not exhibit any significant impact on the selectivity for RebI reduction, it did showcase the most remarkable thermal stability among the mutants tested. Importantly, it maintained higher RebA activity even under elevated temperatures. Most notably, mutant 76_7 demonstrated a 16% higher conversion rate from RebD to RebM. This performance suggests that 76_7 could be effectively employed in industrial applications, particularly in the production of RebM, as we have discussed in the manuscript's corresponding section. These findings collectively underscore the promising potential of these mutants in various applications.

Question 6

The authors need to further polish the manuscript, paying more attention to grammar and typos. For examples:

- 1) Page 6, line 3, rewrite the sentence.
- 2) Page 7, line 17, “63 °C and 66.1 °C”, the decimal places are inconsistent.
- 3) Page 13, line 20, “ $\mu\text{g}\cdot\text{mL}^{-1}$ ” rather than “ $\mu\text{g ml}^{-1}$ ”.
- 4) Page 15, line 21, 2 min rather than 2 minutes in order to be consistent with the description mentioned above.
- 5) Page 15, line 24, “kcat/Km” rather than “kcat/Km” (cat and m should be subscript).

Response:

We appreciate your careful review and specific pointers for the corrections needed in the manuscript. We fully recognize the importance of precise language and consistency for the integrity of our research report. To address your concerns:

- 1) The sentence on Page 6, line 2-4, has been rewritten for clarity.
- 2) On Page 7, line 15, the decimal places for '63 °C and 66.1 °C' have been made consistent.

- 3) The unit on Page 13, line 15, has been corrected to ' $\mu\text{g}\cdot\text{mL}^{-1}$ ' as suggested.
- 4) The time description on Page 15, line 23, has been standardized to '2 min' for consistency.
- 5) The enzyme kinetics notation on Page 16, line 3, has been corrected; 'kcat' and 'Km' are now appropriately subscripted.

Each of these corrections has been meticulously implemented throughout the revised manuscript to ensure textual integrity and consistency.

Reviewer #3 (Remarks to the Author):

This study focuses on employing a Rosetta-based computational design strategy to enhance the thermostability and bioconversion activity of the diterpene glycosyltransferase UGT76G1. Bioinformatics analysis has been used to generate an initial pool of mutations, and Rosetta ddg_monomer was then performed to screen this library. Ultimately, from this collection, ten prime variants were singled out for experimental validation and several of the designed variants exhibited improved thermostability, conversion efficiency and regioselectivity compared to the wildtype. This work provides a computational approach for the rational design of UGT76G1. It is an interesting study and I recommended the publication after the following points properly addressed:

Question 1

On page 5 line 7, mutations with $\Delta\Delta G$ values less than -0.5 kcal/mol were selected. What was the rationale for choosing this specific cutoff value?

Response:

The rationale for choosing a $\Delta\Delta G$ cutoff value of less than -0.5 kcal/mol lies in its empirical basis for assessing protein stability and function. A negative $\Delta\Delta G$ value signifies that a mutation is stabilizing and thus more likely to be tolerated without detrimental impact on the protein's function or structure. The -0.5 kcal/mol threshold is often derived from empirical data suggesting that mutations with $\Delta\Delta G$ values less severe than this are less likely to significantly impact protein stability or functionality. This cutoff serves as a heuristic filter, allowing researchers to focus on mutations that are more likely to be biologically meaningful. Therefore, it acts as an initial screen to narrow down candidates for further empirical or computational validations.

Question 2

Within Figure 1c, the designations for sites N34 and V300 are allocated along the horizontal axis. However, no specific mutants corresponding to these positions are visually represented. Moreover, it is crucial to verify the accuracy of the horizontal axis portrayal in Figure 1c, as it holds the potential to be misleading. For instance, the mutant K52S is situated on the horizontal axis. Yet, in the context of variant information provided in Supplementary Table 1, the K52 position is altered to alanine (A) in

four variants (76_2, 76_3, 76_4, and 76_5). Solely relying on Figure 1c could lead to the erroneous inference that K52S is the exclusive mutant present at position 52 within these five variants. This contrasts with the information in Supplementary Table 1. It is essential to meticulously scrutinize Figure 1c to ensure an accurate representation of mutants, aligning it faithfully with the data in Supplementary Table 1. This measure is pivotal to preempt any potential confusion or misinterpretation arising from inconsistencies between the figure and the supplementary table.

Response:

In response to the reviewer's feedback and suggestion, we have made substantial changes to the presentation of Figure 1c. In the revised figure, we have indicated the mutation sites within the two main domains of the protein.

We acknowledge that mutation K52 cannot be labeled K52S because it changes to alanine (A) in four variants (76_2, 76_3, 76_4 and 76_5). In addition to this, the data in Figure 1c and Supplementary Table 1 were meticulously compared and aligned to ensure no discrepancies to ensure accurate representation of mutations.

Especially, the depiction of the horizontal axis in Figure 1c can be misleading. Therefore, we have described the entire scale of the sequence, and displayed the major components from N-terminal to C-terminal domain. Furthermore, we have meticulously matched every mutation in Supplementary Table 1 to Figure 1c. to eliminate the possibility to confusion caused by just displaying a limited number of mutations. To enhance reader comprehension, we have highlighted the overlapping mutations found in more than 5 variants and the mutations in the substrate binding region.

We believe that these changes will provide readers with a clearer and more easily understandable depiction of the mutation occurrences. We would like to extend our gratitude for the valuable input by the reviewer, which has undeniably improved the quality of our work.

Question 3

On page 6, line 21, it is stated that the L101F mutant is an exclusive feature of variant 76_4. Nonetheless, according to Supplementary Table 1, the L101F mutant is also a constituent of variant 76_2.

Response:

Thank you for your comment, and we appreciate your attention to detail. We apologize for any confusion caused by our previous statement. To clarify, while the L101F mutation is indeed present in both variant 76_4 and variant 76_2, it's important to note that not all mutants containing the L101F mutation exhibit elevated activity. In response to your observation, we have revised the manuscript accordingly. Specifically, we have omitted the sentence on page 6, line 19-20, which previously mentioned the exclusivity of the L101F mutant in variant 76_4. Your feedback has contributed to improving the accuracy of our manuscript, and we thank you for your valuable input.

Question 4

On page 8 line 19, the variant 76_8 exhibits better thermostability and increased T_m values compared to variants 76_6 and 76_7 based on Figure 3a. This raises the question of why variant 76_8 was not

selected over 76_6 and 76_7, given its superior thermal properties.

Response:

As the reviewer mentioned, Variant 76_8 shows better thermal stability and increased T_m value compared to variants 76_6 and 76_7 based on Figure 3a. But, when we conducted activity screening of UGT for 9 mutations that were soluble (extended fig 1), then variant 76_8 failed to produce RebA. Based on this experimental rationale, we did not select variant 76_8. (Page 8, Line 18-19)

Question 5

On page 16 line 2, the software SeeSAR is mentioned, but no citation is provided.

Response:

We appreciate the careful review by the reviewer. Citation is now provided. (Page 16, Line 6-7)

REVIEWERS' COMMENTS:

Reviewer #1 (Remarks to the Author):

Thank you for addressing all my comments. I recommend the acceptance of the manuscript after revising several typo mistakes.

This reviewer would like to ask the authors to check the typo mistakes mentioned throughout the manuscript.

1. Page 2, line 7, T_m rather than Tm. (m should be subscript)
2. Page 9, line 15, “3.6 minutes” should be “3.6 min”. This issue has been mentioned for the first review.
3. Page 13, line 15, “100 $\mu\text{g}\cdot\text{mL}^{-1}$ ” rather than “100 $\mu\text{g}\cdot\text{mL}^{-1}$ ”. (-1 should be both superscript)
4. Page 14, line 7, “18 hours” is the same as issue 2.
5. Page 14, line 19, “25-400 μM ” rather than “25-400uM”. (space and the use of μ)
6. Page 14, line 20, “60 mM” rather than “60mM”. (also space)
7. Page 14, line 23, “ (R^2) ” rather than “(R2)”. (2 should be both superscript)
8. Page 20, line 24, “green.” rather than “green .”. (correct use of full stop)
9. Fig 2, footnote, “76_7 in red.” rather than “76_7 in red)”. (full stop is missing)
10. Fig 5c, the units in the table is not consistent with those in the manuscript. The authors should check and revise them.

To conclude, the authors still need to carefully check the grammar and typo mistakes before acceptance.

Reviewer #3 (Remarks to the Author):

My concerns have been properly addressed and I recommend the publication of this manuscript.

Overall response to reviewer

We appreciate your thorough review of the manuscript and your careful attention to detail. Despite recognizing the importance of accurate language and consistency for the integrity of the research report, identified grammar and typographical errors in the revised manuscript. Each of these corrections has been implemented throughout the revised manuscript to ensure textual integrity and consistency.

Reviewer #1:

Thank you for addressing all my comments. I recommend the acceptance of the manuscript after revising several typo mistakes. This reviewer would like to ask the authors to check the typo mistakes mentioned throughout the manuscript.

1. Page 2, line 7, T_m rather than T_m . (m should be subscript)
2. Page 9, line 15, “3.6 minutes” should be “3.6 min”. This issue has been mentioned for the first review.
3. Page 13, line 15, “100 $\mu\text{g}\cdot\text{mL}^{-1}$ ” rather than “100 $\mu\text{g}\cdot\text{mL}^{-1}$ ”. (-1 should be both superscript)
4. Page 14, line 7, “18hours” is the same as issue 2.
5. Page 14, line 19, “25-400 μM ” rather than “25-400uM”. (space and the use of μ)
6. Page 14, line 20, “60 mM” rather than “60mM”. (also space)
7. Page 14, line 23, “ (R^2) ” rather than “(R2)”. (2 should be both superscript)
8. Page 20, line 24, “green.” rather than “green .”. (correct use of full stop)
9. Fig 2, footnote, “76_7 in red.” rather than “76_7 in red)”. (full stop is missing)
10. Fig 5c, the units in the table is not consistent with those in the manuscript. The authors should check and revise them.

To conclude, the authors still need to carefully check the grammar and typo mistakes before acceptance.

Response:

1. The unit on Page 2, line 7, has been corrected to “ T_m ”.
2. The time description on Page 9, line 15, has been standardized to “3.6 min” for consistency.
3. The unit on Page 13, line 15, has been corrected to “100 $\mu\text{g}\cdot\text{mL}^{-1}$ ” as suggested.
4. The time description on Page 14, line 7, has been standardized to “18 h” for consistency.
5. The unit on Page 14, line 19, has been corrected to “25-400 μM ”
6. The unit on Page 14, line 20, has been corrected to “60 mM”
7. On Page 14, line 23, has been corrected; “ (R^2) ” are now appropriately subscripted.
8. On Page 20, line 24, has been corrected; “green.” are now appropriately indicated.
9. Fig 2, footnote, has been corrected; “76_7 in red.” are now appropriately indicated.
10. Fig 5c, the units in the table has been now consistent with those in the manuscript.